# Stability Analysis of a Patchy Predator–Prey Model with Fear Effect

Tingting Liu and Lijuan Chen *

Center for Applied Mathematics of Fujian Province, School of Mathematics and Statistics, Fuzhou University, Fuzhou 350108, China
* Correspondence: chenlijuan@fzu.edu.cn

**Abstract:** In this paper, a predator–prey model with fear effect and dispersal is proposed. Assume that only the prey migrates at a constant rate between patches and the migration of prey on each patch is faster than the time scale of local predator–prey interaction. Using two time scales, an aggregation system of total prey density for two patches is constructed. Mathematical analysis shows that there may exist a trivial, a boundary and a unique positive equilibrium point. Under certain conditions, the corresponding unique equilibrium point is global asymptotically stable. The impact of the fear effect on the system is also investigated, i.e., the predator density decreases when the amount of fear effect increases. Moreover, dispersal has a great impact on the persistence of the predator and the prey. Numerical experiments are also presented to verify the feasibility of our conclusion.

**Keywords:** patchy model; fear effect; dispersal; stability; persistence

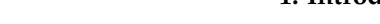

## 1. Introduction

Predator-prey system [1] is one of the most basic relationships among populations, which plays an important role in maintaining the sustainable survival of populations, protecting biodiversity and preventing biological invasion. The predator–prey system with spatial heterogeneity has more complex dynamic properties and rich dynamic behaviors. Environmental heterogeneity can provide spatial shelter for the prey population. The population interaction model in a patchy environment is an effective method to study environmental heterogeneity [2–5]. Thus, the dynamic properties of the patchy predator–prey system have been widely studied [6–9]. Additionally, two time scales are often considered: a fast one in which migrations between patches occur, and a slow one in which interactions, reproduction, and mortality take place. In [10–14], by use of perturbation theory, the full system is aggregated into a reduced system for the total prey and predator populations. For example, Liu [14] proposes a two-patch Leslie–Gower predator–prey model with prey refuge. It is found that the unique positive equilibrium point of the system is always globally asymptotically stable. The above shows that the prey refuge does not change the stability of the system, though it has an effect on the density of the predators and the prey.

At present, not only the direct killing of the prey by the predator is considered, but also the physiological and psychological effects of the predator on the prey are investigated. For example, Wang et al. [15] firstly establish a predator-prey system with fear effect and the dynamic behavior of the system has become complicated under different functional response functions. Consequently, research in this direction has attracted great attention. One can see [16–22] for details. However, to the best knowledge of the authors, no scholar has studied the patchy model with fear effect under two different time scales. Actually, once the prey is suffering from the fear effect, it often leads to migration between patches. In other words, it is more reasonable to discuss the predator–prey system with fear effect and diffusion. Motivated by the above, in this paper, we assume that the prey in patch 1(2) will migrate to patch 2 due to the fear effect, and the prey in patch 2(1) will not be

killed by predators. However, the prey in patch 2 will die exponentially due to lack of other sources of food. In addition, we suppose that migration between patches is much more rapid than the local interaction in the system. In this paper, we shall investigate the following two-patch predator–prey model with fear effect:

$$\begin{aligned}
\frac{dx_1}{d\tau} &= \alpha x_2 - \beta x_1 + \frac{\varepsilon x_1}{1+ky}(r_1 - b_1 x_1) - \varepsilon a_1 x_1 y, \\
\frac{dx_2}{d\tau} &= \beta x_1 - \alpha x_2 + \varepsilon(-dx_2), \\
\frac{dy}{d\tau} &= \varepsilon(-r_2 + a_2 x_1)y,
\end{aligned} \tag{1}$$

where $x_1$, $y$ is the density of the prey and the predator in patch 1 at time $t$, respectively. $x_2$ represents the density of the prey in patch 2. $r_1, b_1, a_1, d, r_2, a_2, \alpha, \beta$ are all positive constants. Here, $r_1$ represents the intrinsic growth rate of prey in patch 1. $d$ is the death rate of prey in patch 2. $b_1$ is the interspecific competition rate. $a_1$ represents the consumption rate of the predator in patch 1 and $a_2$ is the conversion ratio into fitness. $r_2$ is the death rate of the predator. $\beta(\alpha)$ represents the dispersal rate from patch 1(2) to patch 2(1). $\varepsilon$ is an extremely small dimensionless positive number and represents the ratio of fast time scale to slow time scale. $\tau$ is a fast time variable.

As is known to all, based on perturbation techniques and the application of center manifold theorem, the aggregation method [10–14,23–25] reduces a system with a large number of variables involving different time scales into an aggregated system with few global variables. For the complete system (1), an aggregation method will be applied to obtain a reduced system. The reduced form presents a two-dimensional system of ordinary differential equation, which governs the total prey density and the predator density at slow time scale.

In this paper, we choose the aggregated variables as $x = x_1 + x_2$, $y = y$. Adding the first two equations of system (1), we obtain

$$\begin{aligned}
\frac{dx}{d\tau} &= \varepsilon\left(\frac{x_1}{1+ky}(r_1 - b_1 x_1) - dx_2 - a_1 x_1 y\right), \\
\frac{dy}{d\tau} &= \varepsilon\left(-r_2 + a_2 x_1\right)y.
\end{aligned} \tag{2}$$

Let $\varepsilon = 0$ in system (1); we obtain that the solution of the fast part is $x_1^* = \frac{\alpha x}{\alpha+\beta}$, $x_2^* = \frac{\beta x}{\alpha+\beta}$, and then substitute $(x_1^*, x_2^*)$ into system (2). Thus, we can get the following aggregation system:

$$\begin{aligned}
\frac{dx}{dt} &= \frac{\alpha r_1 x}{(\alpha+\beta)(1+ky)} - \frac{\alpha^2 b_1 x^2}{(\alpha+\beta)^2(1+ky)} - \frac{\beta dx}{\alpha+\beta} - \frac{\alpha a_1 xy}{\alpha+\beta}, \\
\frac{dy}{dt} &= \left(-r_2 + \frac{\alpha a_2}{\alpha+\beta}x\right)y,
\end{aligned} \tag{3}$$

where $t = \varepsilon\tau$.

As was pointed out in [10–14,24,25], the complete model is topologically equivalent to the reduced model, and it gives a good approximation solution of the complete model by using the aggregated model. Thus, in the following sections, we will consider the dynamic behaviors of the aggregated model (3).

In order to simplify system (3), we take the following transformations

$$\tau = lt, \ \bar{x} = gx, \ \bar{y} = hy,$$

where

$$l = r_2, g = \frac{a_2 m}{r_2}, h = \frac{a_1 m}{r_2} \ m = \frac{\alpha}{\alpha+\beta}.$$

We still reserve $t$, $x$, $y$ to express $\tau$, $\bar{x}$, $\bar{y}$, respectively. Then, we can obtain the system as follows.

$$\frac{dx}{dt} = \frac{ex}{a+y} - \frac{cx^2}{a+y} - fx - xy,$$
$$\frac{dy}{dt} = y(x-1), \tag{4}$$

where

$$a = \frac{a_1 m}{kr_2}, \quad e = \frac{r_1 a_1 m^2}{kr_2{}^2}, \quad c = \frac{b_1 a_1 m^2}{kr_2 a_2}, \quad f = \frac{(1-m)d}{r_2}.$$

It is not difficult to verify that for system (4), under non-negative initial values, i.e., $x(0) \geq 0, y(0) \geq 0$, the solution $(x(t), y(t))$ is non-negative. Moreover, let $w(t) = x(t) + y(t)$. Suppose that $0 < \sigma < \max\{1, \, f\}$. Then, we have $\frac{dw(t)}{dt} + \sigma w(t) \leq \frac{x}{a+y}(e - cx) \leq \frac{e^2}{4ac}$. Thus, the solution $(x(t), y(t))$ is ultimately bounded.

The rest of the paper is arranged as follows. In Section 2, the existence and stability of the equilibria are established. In Section 3, we show the global stability of the equilibria. In Section 4, we analyze the impact of fear effect and dispersal on predator density and give some numerical simulation results to verify these analytical results. We summarize the obtained theoretical results and biological application in Section 5.

## 2. Existence and Local Stability of Equilibrium

In this section, we shall discuss the existence and stability of the equilibrium for system (4). The equilibrium of system (4) is given by the following equation:

$$\begin{cases} \dfrac{ex}{a+y} - \dfrac{cx^2}{a+y} - fx - xy = 0, \\ y(x-1) = 0. \end{cases}$$

By simple computation, there always exist a trivial equilibrium $E_0(0,0)$ and a boundary equilibrium $E_1(\frac{e-af}{c}, 0)$. To obtain the positive equilibrium for the system, we consider the following equation:

$$y^2 + (a+f)y + af + c - e = 0. \tag{5}$$

Let the discriminant of Equation (5) denoted by $\Delta = (a+f)^2 - 4(af + c - e)$. If $e > af + c$, then $\Delta > 0$ and system (4) has a unique positive equilibrium $E_2(x^*, y^*)$. Here, $x^* = 1$ and $y^* = \frac{-(a+f)+\sqrt{\Delta}}{2}$.

Next, we will investigate the stability of the above equilibrium. The Jacobian matrix of system (4) at $E(x,y)$ is calculated as

$$J_E = \begin{bmatrix} \dfrac{e-2cx}{a+y} - f - y & x\left(\dfrac{cx-e}{(a+y)^2} - 1\right), \\ y & -1+x. \end{bmatrix}$$

The dynamics of system (4) are given as follows.

**Theorem 1.**

(1) System (4) always has a trivial equilibrium $E_0(0,0)$.

    (a)    If $e < af$, $E_0(0,0)$ is a hyperbolic stable node;
    (b)    If $e = af$, $E_0(0,0)$ is an attracting saddle-node;
    (c)    If $e > af$, $E_0(0,0)$ is a hyperbolic saddle.

(2) If $e > af$, system (4) has a boundary equilibrium $E_1(x_1, 0)$. Here, $x_1 = \frac{e-af}{c}$.

    (a)    If $af < e < af + c$, $E_1(x_1, 0)$ is a hyperbolic stable node;
    (b)    If $e = af + c$, $E_1(x_1, 0)$ is an attracting saddle-node;
    (c)    If $e > af + c$, $E_1(x_1, 0)$ is a hyperbolic saddle.

(3)　When $e > af + c$, system (4) has a unique positive equilibrium $E_2(x^*, y^*)$, which is always asymptotically stable.

**Proof.** (1) The Jacobin matrix at $E_0(0,0)$ is

$$J_{E_0} = \begin{bmatrix} \frac{e}{a} - f & 0 \\ 0 & -1 \end{bmatrix}.$$

By direct calculation, we can obtain two eigenvalues of $J_{E_0}$, i.e., $\lambda_1 = \frac{e}{a} - f$ and $\lambda_2 = -1 < 0$. If $e > af$, $E_0(0,0)$ is a hyperbolic saddle. If $e < af$, we have $\lambda_1 < 0$, then $E_0(0,0)$ is a hyperbolic stable node. If $e = af$, then $\lambda_1 = 0$. In order to investigate the stability property of the equilibrium $E_0$, we introduce the new time variable $\tau = \frac{-a}{a+y}t$, and then we have

$$\frac{dx}{d\tau} = \left(1 + \frac{f}{a}\right)xy + \frac{c}{a}x^2 + \frac{1}{a}xy^2,$$
$$\frac{dy}{d\tau} = y - xy - \frac{1}{a}xy^2 + \frac{1}{a}y^2,$$

the coefficient of $x^2$ is $\frac{c}{a} > 0$. Considering the new time variable $\tau$ and using Theorem 7.1 in [26], we can conclude that $E_0$ is an attracting saddle-node.

(2) The Jacobian matrix at $E_1(x_1, 0)$ is

$$J_{E_1} = \begin{bmatrix} \frac{e - 2cx_1}{a} - f & x_1\left(\frac{cx_1 - e}{a^2} - 1\right) \\ 0 & -1 + x_1 \end{bmatrix}.$$

By simple calculation, we get two eigenvalues of $J_{E_1}$, i.e., $\lambda_1 = \frac{af - e}{a} < 0$, $\lambda_2 = -1 + x_1$. It follows that if $af < e < af + c$, $\lambda_2 < 0$, and so $E_1$ is a hyperbolic stable node. If $e > af + c$, then $E_1$ is a saddle. When $e = af + c$, it follows that $\lambda_2 = 0$. To analyze the stability of $E_1(x_1, 0)$, we make the transformation $(X, Y) = (x - x_1, y)$ and expand the power series around the origin. Then, system (4) becomes

$$\frac{dX}{dt} = -\frac{c}{a}X - \frac{f + a}{a}Y - \frac{c}{a}X^2 - \frac{a^2 + af - c}{a^2}YX + \frac{f}{a^2}Y^2 + P_1(X, Y),$$
$$\frac{dY}{dt} = XY.$$

Here, $P_1(X, Y)$ denote the power series with term $U^i V^j$ satisfying $i + j \geq 3$.

Let $U = X + \frac{f+a}{c}Y$, $V = Y$. Introducing a new time variable $\tau$ by $\tau = \frac{-c}{a}t$ and rewriting $\tau$ as $t$, we have

$$\frac{dU}{dt} = U + \frac{1}{a}U^2 - \frac{af(f - a + 1) - f^2 - c}{a^2c}V^2 - \frac{a^2 + af + c}{ac}UV + P_2(U, V),$$
$$\frac{dV}{dt} = UV - \frac{f + a}{c}V^2,$$

where $P_2(U, V)$ denote the power series with term $U^i V^j$ satisfying $i + j \geq 3$.

Hence, by Theorem 7.1 in Chapter 2 in [26], the coefficient of $V^2$ is $-\frac{f+a}{c} < 0$, then the equilibrium $E_1$ is an attracting saddle node.

(3) The Jacobian matrix at $E_2$ is

$$J_{E_2} = \begin{bmatrix} \frac{e - 2c}{a + y^*} - f - y^* & \frac{c - e}{(a + y^*)^2} - 1 \\ y^* & 0 \end{bmatrix}.$$

Under the existence condition of $E_2$, i.e., $af + c < e$, it is easy to obtain that $Det(E_2) = -y^*(\dfrac{c-e}{(a+y^*)^2} - 1) > 0$ and $Tr(E_2) = \dfrac{e-2c}{a+y^*} - f - y^* = \dfrac{-c}{a+y^*} < 0$. Thus, $E_2(x^*, y^*)$ is locally asymptotically stable. □

The corresponding illustration of Theorem 1 is shown in Figure 1 by matlab software.

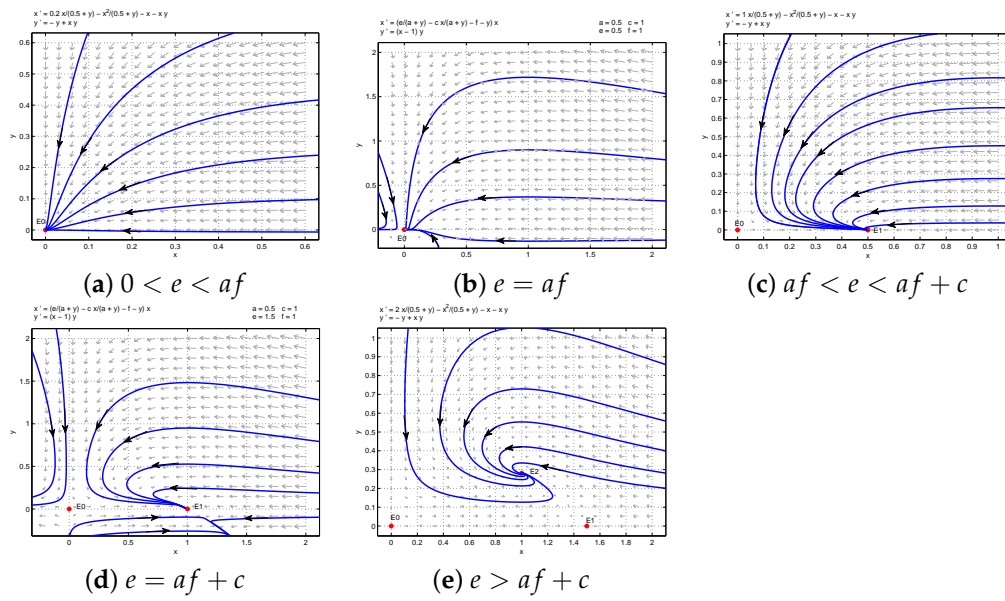

**Figure 1.** The phase portraits of system (4).

## 3. Global Stability of Equilibrium

In this section, for system (4), we will investigate the global stability of equilibria.

**Theorem 2.** *If $e < af$, $E_0(0,0)$ is globally asymptotically stable.*

**Proof.** When $e < af$, system (4) has a unique stable equilibrium $E_0(0,0)$. The solution of system (4) is bounded, so the solution cannot extend to the infinity. Additionally, there exists no limit cycle, since system (4) has no interior equilibrium. Thus, $E_0(0,0)$ is globally asymptotically stable. □

**Theorem 3.** *If $af < e < af + c$, $E_1(x_1, 0)$ is globally asymptotically stable.*

The proof of Theorem 3 is similar to that of Theorem 2; thus, we omit it for brevity.

**Theorem 4.** *If $e > af + c$, $E_2(x^*, y^*)$ is globally asymptotically stable.*

**Proof.** Here, we consider the Dulac function $B(x, y) = \dfrac{1}{xy}$. Then,

$$\frac{\partial(BP)}{\partial x} + \frac{\partial(BQ)}{\partial y} = \frac{-c}{(a+y)y} < 0,$$

where

$$P = \frac{ex}{a+y} - \frac{cx^2}{a+y} - fx - yx,$$
$$Q = -y + xy.$$

According to the Bendixson–Dulac discriminant, system (4) has no limit cycle in the first quadrant. Thus, $E_2(x^*, y^*)$ is globally asymptotically stable. The proof of Theorem 4 is finished. □

Notice that in Theorems 2–4, the conditions $e < af$, $af < e < af + c$ and $e > af + c$ are equivalent to $r_1 < \frac{\beta}{\alpha}d$, $\frac{\beta}{\alpha}d < r_1 < \frac{\beta}{\alpha}d + \frac{b_1 r_2}{a_2}$ and $r_1 > \frac{\beta}{\alpha}d + \frac{b_1 r_2}{a_2}$, respectively. Thus the intrinsic growth rate of the prey in patch 1 plays an important role in the sustainable development of the species. In detail, when $r_1$ is not large, both the prey and the predator are led to extinction. Meanwhile, intermediate intrinsic growth rate $r_1$ can be favorable to the survival of the prey. However, an extremely large intrinsic growth rate could result in the coexistence of both the prey and the predator.

## 4. The Impact of Fear Effect and Dispersal

### 4.1. The Impact of Fear Effect on the Predator Density

Next, we will discuss the impact of the fear effect on the predator density. Computing the derivation along the predator $y^*$ with respect to $k$, we have

$$
\begin{aligned}
\frac{dy^*}{dk} &= \frac{2a\sqrt{\Delta} - (2(a-f)a - 4(c-e))}{4k\sqrt{\Delta}} \\
&= \frac{f^2 - (\sqrt{\Delta} - a)^2}{4k\sqrt{\Delta}}.
\end{aligned}
$$

Under the existence condition of $E_2$, i.e., $e > af + c$, one can obtain $\sqrt{\Delta} - a > f$. Thus, we can easily deduce $\frac{dy^*}{dk} < 0$. Additionally,

$$
\lim_{k \to \infty} \frac{dy^*}{dk} = 0
$$

which shows that $y^*$ is a strictly decreasing function of $k$, i.e., increasing the fear effect $k$ may decrease the predator density.

### 4.2. The Impact of Dispersal on the Predator Density

The derivation of $y^*$ with respect to $m$ along system (4) is as follows.

$$
\begin{aligned}
\frac{dy^*}{dm} &= \frac{(1-m) \cdot \left[-a\sqrt{\Delta} + (a-f)a - 4(c-e)\right]}{2\sqrt{\Delta} \cdot m \cdot (1-m)} \\
&+ \frac{fm(\sqrt{\Delta} + a - f)}{2\sqrt{\Delta} \cdot m \cdot (1-m)}.
\end{aligned}
$$

Under the existence condition of $E_2$, i.e., $e > af + c$, one can get $\sqrt{\Delta} > a + f$. Additionally, notice that: $-a\sqrt{\Delta} + (a-f)a - 4(c-e) > f(\sqrt{\Delta} + a - f) > 0$. Based on the above analysis, we can get $\frac{dy^*}{dm} > 0$. In other words, $y^*$ increases as $m$ increases.

Figure 2a,b show how fear effect and dispersal have a direct influence on the predator density, respectively. In other words, the predator density decreases when $k$ increases or $m$ decreases. In fact, it is not difficult to understand the above observation biologically. When the fear effect $k$ increases or $m$ decreases, the prey in patch 1 is more likely to move to refuge patch 2, which will lead to a decease in the total prey density. Consequently, the predation density decreases due to the reduced amount of food.

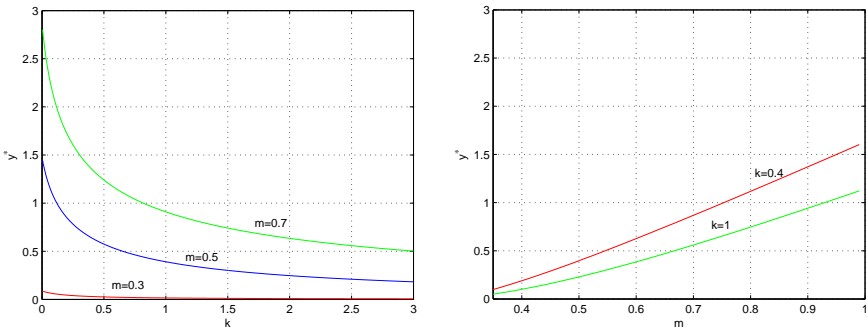

(**a**) *The relation between $y^*$ with k.*    (**b**) *The relation between $y^*$ with m.*

**Figure 2.** The total predator density $y^*$ with respect to $k$ and $m$.

## 5. Conclusions

In this work, a patchy predator–prey system with a fear effect has been proposed and analyzed. Only the prey population migrates at a constant rate between patches, and there are two time scales, i.e., a fast one for migration of prey between patches, and a slow one corresponding to local predator–prey interaction. The aggregated model is studied analytically, and the threshold conditions for the existence and stability of various steady states are worked out in Theorems 1–4. In detail, the fear effect does not change the stability of the unique positive equilibrium of the system, which is in agreement with [15]. However, unlike [15], dispersal has a vital influence on the species' survival. In fact, when $m < \frac{d}{r_1+d}$, i.e., $\beta > \frac{r_1}{d}\alpha$, both the prey and the predator species are led to extinction. When $\frac{d}{r_1+d} < m < \frac{d}{r_1+d-\frac{b_1 r_2}{a_2}}$, i.e., $\frac{r_1}{d}\alpha < \beta < \frac{r_1-\frac{b_1 r_2}{a_2}}{d}\alpha$, the prey species is permanent, but the predator species is led to extinction. When $m > \frac{d}{r_1+d-\frac{b_1 r_2}{a_2}}$, i.e., $\beta > \frac{r_1-\frac{b_1 r_2}{a_2}}{d}\alpha$, both the prey and the predator can persist. From the above, whether the species can survive or not depends on the dispersal. The results show that the dispersal and fear effect play an important role in the dynamic behaviours of the system, which is a good extension and supplements those in [14,27]. We would also like to point out here that several potential directions can be investigated. For example, in this paper, we only discuss two patches for the prey. It is also interesting to propose a model with dispersal among $n > 2$ patches. Moreover, similar to [28–30], once there is noise or the conformable derivative in the system, respectively, how about the corresponding dynamic behaviour? We leave this for future work.

**Author Contributions:** Writing—original draft preparation, T.L.; writing—review and editing, L.C. All authors have read and agreed to the published version of the manuscript.

**Funding:** This work was supported by the National Natural Science Foundation of China under Grant (11601085) and the Natural Science Foundation of Fujian Province (2021J01614, 2021J01613).

**Institutional Review Board Statement:** Not applicable.

**Informed Consent Statement:** Not applicable.

**Data Availability Statement:** Not applicable.

**Acknowledgments:** The authors would like to thank the anonymous reviewers for their valuable comments, which greatly improved the final expression of the paper.

**Conflicts of Interest:** The authors declare that there are no conflict of interest.

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
