# Peer review of "Stability Analysis of a Patchy Predator–Prey Model with Fear Effect"

_axioms, doi:10.3390/axioms11100577_

Round 1
Reviewer 1 Report
I feel myself conecrned with the following issues, which I put forward for improving the new edition.
1. The `aggregation' technique is widely known for a very long time as the multiple scale expansion method. The variant in use seems to go back to Bogolyubov. The reference provided is misleading, therefore.
2. The statement regarding the topological equivavelnce between the original and aggregated systems is unclear and likely incorrect. Of course, it does not mean that considering the latter does not make sence.
3. `The fear effect' is likely had been studied before in the context of Keller-Siegel models e.g., see Pursuit-evasion predator-prey waves in two spatial dimensions by V. N. Biktashev et al.
4. I was unable to see what do the authors mean by `diffusion'. I did not see any.
5. Propositions on the global stability of the equilibria need more clear proofs of the global boundedness of the positive solutions.
Reviewer 3 Report
#Please consider the attached file.

Reviewer 4 Report
The article is devoted to an important issue - stability analysis of a patchy predator-prey model with fear effect.
The authors propose a predator-prey model with fear effect and diffusion. They also investigate the impact of fear effect on the system and observe that the predator density decreases when the amount of fear effect increases. They find that diffusion has a great impact on the persistence of the predator and prey. Numerical experiments further verify the feasibility of our conclusions.
Introduction is rather good, but in the end of this section phrase «We end this paper by Section 5» can be improved. it is advisable to indicate the thesis content of this section by analogy with the text presented above in this paragraph.
Reviewer 5 Report
Review Report:
I believe that the results of this manuscript are interesting, but some issues should be revised before the manuscript proceeds for publication.
1. The authors should write the abstract as passive. Using ‘’we’’ is not recommended.
2. There is no information regarding the solution method in the introduction. A detailed review is recommended.
3. Which package was used to handle the computations? It should be mentioned in the manuscript.
4. The conclusion should be written as the past.
5. The references list should be updated as there are various methods to handle such a model like ‘’The guava model involving the conformable derivative and its mathematical analysis’’.
After addressing the above suggestions, I recommend the current manuscript for publication.
Round 2
Reviewer 1 Report
I'm not happy with the references you had provided regarding the issue of topological equivalence. I do not believe that this work delivers a proof the completness and accuracy of which are in line with the mathematical understanding of what these matters are.
Author Response
Thanks for your careful reading. As we know, lots of literatures have
pointed out that the aggregation method can reduces a system with large number of variables into an aggregated system with few global variables. The reduced form presents a two dimensional system of ordinary differential equation which governs the total prey density and the predator density at slow time scale. For example, as was pointed out in Page 127 of Ref.[12], model (3) can be obtained by adding the first two equations of model (2). Then substituting x_1* and x_2* into model (3), one can obtain the aggregated system (4) at the slow time scale. The authors of Ref,[12] consider the dynamic behaviors of model (4) and show the topological equivalence between model (2) and model (4).
Reviewer 2 Report
The authors have slightly improved the paper, even though it is not of high quality.
Author Response
Thanks. We have tried our best to check the whole paper carefully.
And we hope that the revised manuscript can be more suitable to be published.
Reviewer 3 Report
#Please consider the attached file.

Author Response
Thanks for your valuable suggestion. We have checked all the above
and have tried our best to revise the whole paper carefully.